# Exposure to the RXR Agonist SR11237 in Early Life Causes Disturbed Skeletal Morphogenesis in a Rat Model

**DOI:** 10.3390/ijms20205198

**Published:** 2019-10-20

**Authors:** Holly Dupuis, Michael Andrew Pest, Ermina Hadzic, Thin Xuan Vo, Daniel B. Hardy, Frank Beier

**Affiliations:** 1Departments of Physiology and Pharmacology, Schulich School of Medicine and Dentistry, Western University London, London, ON N6A 5C1, Canada; hkipp@uwo.ca (H.D.); mpest@uwo.ca (M.A.P.); ehadzic2@uwo.ca (E.H.); Thinxv@gmail.com (T.X.V.); Daniel.Hardy@schulich.uwo.ca (D.B.H.); 2Western Bone and Joint Institute, Western University London, London, ON N6A 5C1, Canada; 3Obstetrics and Gynecology, Schulich School of Medicine and Dentistry, Western University London, London, ON N6A 5C1, Canada

**Keywords:** osteoarthritis, retinoid X receptor, endochondral ossification, chondrocytes

## Abstract

Longitudinal bone growth occurs through endochondral ossification (EO), controlled by various signaling molecules. Retinoid X Receptor (RXR) is a nuclear receptor with important roles in cell death, development, and metabolism. However, little is known about its role in EO. In this study, the agonist SR11237 was used to evaluate RXR activation in EO. Rats given SR11237 from post-natal day 5 to post-natal day 15 were harvested for micro-computed tomography (microCT) scanning and histology. In parallel, newborn CD1 mouse tibiae were cultured with increasing concentrations of SR11237 for histological and whole-mount evaluation. RXR agonist-treated rats had shorter long bones than the controls and developed dysmorphia of the growth plate. Cells invading the calcified and dysmorphic growth plate appeared pre-hypertrophic in size and shape, in correspondence with p57 immunostaining. Additionally, SOX9-positive cells were found surrounding the calcified tissue. The epiphysis of SR11237-treated bones showed increased TRAP staining and additional TUNEL staining at the osteo-chondral junction. MicroCT revealed morphological disorganization in the long bones of the treated animals. This study suggests that stimulation of RXR causes irregular ossification, premature closure of the growth plate, and disrupted long bone growth in rodent models

## 1. Introduction

Longitudinal bone growth occurs through the process of endochondral ossification, in which a cartilaginous model first expands through the activity of chondrocytes and then is replaced by bone. Endochondral ossification begins with the condensation of mesenchymal cells that subsequently differentiate into chondrocytes, the primary cell type of cartilage. Chondrocytes begin expressing SRY (sex-determining region Y)-box9 (SOX9) and synthesize a matrix comprised largely of type II collagen and the proteoglycan aggrecan [1]. These early cartilage cells have a distinct rounded shape, are scattered irregularly throughout the matrix, divide infrequently, and give rise to resting chondrocytes. Eventually, the resting cells begin to divide more rapidly and enter the proliferative zone, although the driving force behind this shift in activity is not completely known [2]. While in the proliferating zone, cells appear flattened. Upon division, their daughter cells line up in columns along the longitudinal axis of the bone. The elongated shape and the longitudinal growth of endochondral bones are largely due to this spatial orientation [2], with chondrocyte proliferation as one key factor influencing longitudinal bone growth. Chondrocytes in the center of the cartilage eventually cease to proliferate and exit the cell cycle to become hypertrophic cells. Chondrocytes undergoing hypertrophy increase up to 20-fold in size [3], stop synthesizing SOX9, and begin expressing type X collagen instead of type II. Hypertrophic differentiation is driven by transcription factors such as runt-related transcription factor 2 (RUNX2) [4,5,6] and myocyte enhancer factor-2c (MEF2C) [7] that is required for Runx2 expression [1]. Hypertrophic chondrocytes release signals to attract blood vessels, osteoclasts invade the matrix to resorb the cartilage, and trabecular bone formation initiates. During this process, hypertrophic chondrocytes either undergo apoptosis or trans-differentiate into osteoblasts [1,8,9]. Processes occur in the center of the cartilage template, forming the primary ossification center. A secondary ossification center forms later at the end of the template through a similar process. The growth plate consists of the chondrocytes located between these two areas of ossification, with distinctly divided zones of resting, proliferating, and hypertrophic cells [10]. Throughout this process, proliferation and hypertrophy of chondrocytes drive the longitudinal growth of the bone [11]. A complex array of hormones and signaling molecules control the process of endochondral ossification [12]. Dysregulation of any of the factors controlling growth plate function can cause defects including reduced bone length, altered mineralization, and skeletal deformities, such as seen in chondrodysplasias [1,5,10,11,12]. Understanding these processes is crucial to elucidating the etiology of bone diseases along with developing interventions to ameliorate any adverse outcomes.

Retinoid X Receptor (RXR) is a type 2 nuclear receptor that is part of the steroid/thyroid hormone superfamily. It has important roles in cell death, development, metabolism, and cell differentiation [13] and is expressed in almost every tissue of the body [14]. There are three different RXR proteins (alpha, beta, and gamma), and each is expressed in a cell type-dependent manner [13]. RXR has the ability to form homodimers or heterodimers with other nuclear receptors in a permissive or non-permissive way [15,16]. Permissive heterodimers can be activated by agonists for either partner in the dimer and include RXR dimers with peroxisome proliferator-activated receptors (PPARs), farnesoid X receptors (FXRs), and liver X receptors (LXRs). Non-permissive dimers can only be activated by the ligand for the partner (not by RXR ligands) and include retinoic acid receptors (RARs), vitamin D receptors (VDRs), and thyroid hormone receptors (TRs). Given that RXR homodimerizes with itself and heterodimerizes with several nuclear receptors, thereby regulating the downstream transcription of multiple genes, the direct influence of RXR alone on development has been difficult to elucidate. The RXR binding partner most broadly studied in relation to rodent development is RAR. Although this complex is important for many processes in development, the most profound effects of RXR–RAR heterodimers have been discovered in embryonic bone. For example, exposure to retinoic acid during embryonic development causes skeletal element deletions and truncations in the forelimbs [17]. While RXR is involved in these processes, the RXR–RAR heterodimer is primarily activated by an RAR agonist [18]. It remains to be studied how RXR homodimers would affect bone development without the influence of any other binding partner. Although the impact of RXR itself is far-reaching, little is known about its relevance in long bone development. To determine its importance, the RXR-specific pan-agonist SR11237 [19] was used in this study to identify the effects of RXR activation on rat endochondral bone development.

## 2. Results

### 2.1. RXR Activation Results in Reduced Body Weight and Bone Length

Newborn rats were injected with DMSO or SR11237 daily from P5 to P15. Daily treatment with vehicle or SR11237 had no effects on maternal behavior or neonatal mortality. At P16, the animals were weighed before sacrifice. The RXR agonist-treated animals of both sexes weighed 30% less than the DMSO-injected controls (Figure 1A,B). Humerus, tibia, femur, and radius were dissected from the right side of all Sprague-Dawley pups and measured for length. A non-parametric Mann–Whitney unpaired two-tailed t-test showed that the femur, tibia, and radius bones were significantly shorter in the females injected with SR11237 than in the DMSO controls (Figure 1C,D). All four bones were significantly shorter in the males injected with SR11237 than in the DMSO controls (Figure 1C,D).

### 2.2. The RXR Agonist SR11237 Decreases Total Length Change in Cultured Mouse Tibiae

Tibiae were isolated from newborn mice and cultured for 4 days with DMSO (control) or differing concentrations of the RXR agonist SR11237. The total length of bones was measured at the beginning and end of the culture period to determine the longitudinal growth (Figure 2). Treatment of tibiae with 5 μM SR11237 (Figure 2A) caused a decrease in length in comparison to all other conditions, but statistical significance was only observed in comparison to the 1 μM treatment. Safranin O/Fast Green staining of tibia sections confirmed that DMSO-treated tibiae were longer than tibiae treated with 5 μM SR11237 (Figure 2B).

### 2.3. MicroCT Analyses Shos Abnormal Bone Morphology in RXR Agonist-Treated Rats

The left limbs of P16 male rats treated with RXR agonist or control were harvested for Micro-computed tomography (microCT) analysis. The shape and size of all long bones changed substantially upon treatment with the RXR agonist. The images demonstrated a significant size reduction in the fore- and hindlimbs treated with SR11237 compared to the DMSO control-treated limbs (Figure 3A,B). Regions of increased and decreased radio-opacity could be seen in all bones of the fore- and hindlimbs, scapula, hands, and feet upon RXR agonist treatment (Figure 3A–E). These scans indicated that the scapula showed less calcification in the center of the bone, but thicker calcification along the outer rim in the animals treated with the RXR agonist (Figure 3C). Metacarpals and metatarsals of the hands and feet appeared dysmorphic, under-calcified, and under-developed in treated rats (Figure 3D,E). SR11237 treatment resulted in the disruption of the development of the secondary ossification centers, as well as in the reduction of the size of mineralization of the smaller bones of the hands and feet (Figure 3D,E).

### 2.4. Disrupted Growth Plate Morphology in P16 Male Rat Long Bones

Paraffin sections of humerus, tibia, and femur from the P16 rats were stained with Safranin O/fast green (Figure 4). This staining highlighted greatly disturbed growth plate organization and the apparent fusion of the primary and secondary ossification centers in the RXR agonist-treated animals versus DMSO controls. RXR activation by SR11237 caused premature growth plate closure and an infiltration of ossified tissue through the central epiphysis of the bones, effectively joining together the primary and secondary ossification centers. Importantly, the tibia appeared to be more afflicted than the humerus and the femur (Figure 4). Females showed a similar but less severe phenotype than males (Appendix A).

### 2.5. Immunohistochemistry and Histological Staining of P16 Rat Tibial Sections

We performed histological and immunohistological analyses to further characterize the observed phenotypes. SOX9 (a marker for early and proliferating chondrocytes) expression was found in growth-plate chondrocytes in control rats and in the cells surrounding the calcified tissue in SR11237-treated rats (Figure 5). The cartilage of SR11237-treated animals showed reduced numbers of chondrocytes staining positive for PCNA, a marker of cell proliferation, and P57, a marker of post-mitotic and pre-hypertrophic chondrocytes (Figure 5). In contrast, the central epiphysis of SR11237-treated rats was highly positive for TRAP and picro-sirius red staining, suggesting increased osteoclast activity and accelerated replacement of cartilage by bone (Figure 6). TUNEL staining showed concentrated cell death at the osteo–chondral junction of the growth plate in both groups (Figure 6).

## 3. Discussion

This study examines the effects of RXR agonist administration on rodent endochondral ossification and growth plate biology. By activating RXR specifically in the early newborn stage, prior to and during the onset of secondary ossification, using the RXR agonist SR11237, we found adverse effects on the growth plate of rat long bones, as well as effects on body size, bone formation, and morphology of multiple skeletal elements. Previous studies had examined the effects of other RXR agonists on bone parameters but were largely focused on adult stages [20]. Similarly, mice with specific deletions of the RXRα and RXRβ genes in hematopoietic cells showed defects in osteoclast function but were studied at later stages only [21]. The RXR agonist-treated animals of both sexes were significantly smaller (30%) than their littermate DMSO-injected siblings. This size difference was present until sacrifice at P16. Similarly, all four long bones dissected (tibia, femur, humerus, and radius) were significantly shorter in the animals injected with SR11237 compared to DMSO controls (with the exception of the female humerus, which was not significantly shorter). At present it is unclear whether the effects on body weight are due to systemic and metabolic effects or secondary to reduced bone growth. However, the reduced growth of tibiae cultured with the high concentration of the RXR agonist suggests that at least a portion of the growth retardation observed was due to direct effects on growth-plate chondrocytes. The effects of the RXR pan-agonist SR11237 on endochondral ossification was further evaluated in a mouse ex vivo model to further focus on direct effects on the skeleton and validate our findings in a second species. The growth of the mouse long bones was reduced by the treatment with 5 µM SR11237 (at least in comparison to the 1 µM treatment), further emphasizing the profound effects of RXR stimulation during long bone development and growth. The mouse ex vivo experiments were done for a shorter duration (4 days) than the in vivo studies (10 days), providing one potential reason for the somewhat lower effects of SR11237 in this model. Additional possible explanations include age of the animals (P0 versus P10), species differences, and indirect effects of SR11237 through other organs (such as endocrine glands) in the rat model.

MicroCT imaging of the rats demonstrated that the fore- and hindlimbs of the SR11237-injected animals were thinner and appeared somewhat osteopenic. Skeletal staining with Alcian Blue and Alizarin Red revealed similar findings. The metacarpals and metatarsals of the hands and feet appeared dysmorphic, undercalcified, and underdeveloped. Although both sexes appeared to be afflicted, males appeared more impaired than females. Safranin O/ fast green staining of long bone sections highlighted disturbed growth plate organization and the disruption of the primary and secondary ossification centers in the RXR agonist-treated animals versus the DMSO controls. Cells surrounding the disrupted growth plate and ossification centers were largely pre-hypertrophic in size and shape. Again, males appeared to be more affected than females. Importantly, the tibia appeared to be more afflicted than the humerus and the femur. In all staining and immunohistochemical experiments, sections from both male and female rats were used. However, in every instance, males appeared to be more afflicted than females. P57, SOX9, and PCNA immunohistochemistry was performed on tibia, humerus, and femur sections. All three markers were present in control animals in the expected patterns, but the staining patterns in the animals treated with SR11237 further demonstrated the extreme disorganization of the growth plate. Upon RXR activation, PCNA, P57, and SOX9 were found in the cells surrounding the central ossified section in no particular spatial order, suggesting that the overall organized structure of the zones of the growth plate was lost, and chondrocytes of various maturation stages were intermingled. The reduced number of PCNA-positive chondrocytes upon SR11237 administration suggests that RXR activation drives the cells out of the cell cycle, which could contribute to reduced bone growth and to the closure of the growth plate. Although it was difficult to tell if the SR11237-injected animals were more afflicted than their DMSO controls, the pattern of TUNEL staining appeared to be moderately different. Generally, the highest amount of cell death in a typically developing growth plate occurs in the late hypertrophic zone, where some chondrocytes are undergoing apoptosis and osteoclasts are invading [22]. However, in the SR11237-treated animals, TUNEL-positive cells appeared most consistently around the infiltration of ossified tissue through the growth plate, seemingly in no particular order, again demonstrating the intermingling of multiple chondrocyte types in the same space. However, there did appear to be an increase in cell death at the osteo–chondral junction, which is consistent with the normal transition, as some hypertrophic cells undergo apoptosis. The highly localized pattern of TUNEL staining also suggests that the severe phenotypes observed are not due to general cellular toxicity induced by the RXR agonist.

In the DMSO control animal growth plate, TRAP staining occurs only at the site where the hypertrophic cartilage transitions to bone and around the area of secondary ossification. In the RXR agonist-treated animals, TRAP staining was found throughout the central epiphysis of the bone and connected the primary and secondary ossification centers completely. The staining appeared very intense and prevalent throughout, suggesting that osteoclast activation or recruitment is one of the main responses to RXR activation.

Picro-sirius red stains collagen fibers and is particularly strong in bone tissue. Control animals showed picro-sirius staining concentrated at the primary and secondary ossification centers, but only weak staining in the area of the growth plate. Upon RXR activation, the area and intensity of staining appeared greatly increased, further supporting our finding of increased bone formation and highlighting the fusion of primary and secondary ossification centers. Future work will need to address whether the effects outlined in this study are due to RXR action itself or involve any of its heterodimeric partners. Potential candidates include the vitamin D receptor (VDR), since it stimulates the expression of RANKL (receptor activator of nuclear factor kappa-B ligand) in cartilage, resulting in the activation of osteoclasts at the osteochondral junction [23]. In addition, deletion of the gene for RXRγ worsens the growth plate phenotype of VDR knock-out (KO) mice, providing additional evidence for a shared function of these receptors in cartilage [24].

Similarly, retinoic acid (potentially through retinoic acid receptors, RARs) stimulates cartilage-to-bone turnover in vitro [25], resembling some of the phenotypes we observed here. However, since both VDR and RAR are non-permissive partners of RXR, it seems less likely that they mediate the effects of a specific RXR agonist. Mouse knockout studies also suggest reduced osteoclast activity in mice with deletions of the genes encoding several other partners of RXR, including PPARγ, LXRs, and thyroid hormone receptor alpha (reviewed in [26]); thus, activation of the respective heterodimers by our RXR agonists could contribute to the observed effects. However, it should be noted that in parallel experiments, administration of the LXR agonist GW3695 did not induce any obvious bone growth effects. Overall, it seems quite plausible that the complex effects observed in our study could be due to the combined activation of RXR homodimers and several heterodimeric pairs, potentially differently in the different cell types involved (chondrocytes, osteoblasts, osteoclasts, and maybe more).

In conclusion, our studies demonstrated that the administration of the selective RXR agonist SR11237 in skeletally immature rats caused irregular ossification, premature closure of the growth plate, and reduced bone growth during early postnatal development. Consequently, prolonged RXR signaling, especially in early life, may result in long-term effects on long bone and joint morphology and lead to additional pathology (such as osteoarthritis) through joint malalignment, disruption of normal gait, or direct effects on articular chondrocytes.

## 4. Materials and Methods

### 4.1. Animals

Sprague-Dawley rats and pregnant CD1 mice were purchased from Charles River Laboratories (St. Constant, QC, Canada). Animals were housed in a controlled-temperature (20–25 °C), controlled-humidity (40–60%), ventilated animal room. They were bred in-house and given free access to water and standard rat or mouse chow. Once born, mouse pups were euthanized by asphyxiation with CO_2_ and harvested immediately for organ cultures. The rat pups were housed with their mother upon birth until harvest at post-natal day 16 (P16). The rats were euthanized by asphyxiation with CO_2_. All animal experiments were handled in accordance to the guidelines from the Canadian Council on Animal Care and were approved by the Animal Use Subcommittee at the University of Western Ontario (2019-035 (01-06-2019)). 

### 4.2. SR11237 Injections

In each trial, Sprague-Dawley rat pups were divided randomly into two groups. The first group was given an intraperitoneal injection (IP) of SR11237 (Sigma, Oakville, ON, Canada, #S8951) (pan-RXR specific agonist) at a concentration of 25 mg per kg body weight (dissolved in DMSO). The second group was injected with the same volume of DMSO (dimethyl sulfoxide) (Sigma, Oakville, ON, Canada, #472301) vehicle. Neonates were injected IP once a day from post-natal day 5 to post-natal day 15, in a similar regime as published for GLP (glucagon-like peptide) agonists in newborns [27], during a critical window of bone development in the rat [1,5,10,11,12]. This dose of SR11237 is similar to the one previously used in vivo [28]. The animals were harvested on post-natal day 16. The animals were weighed at time of harvest.

### 4.3. Tibia Organ Cultures

Tibiae were harvested from newborn CD1 mice and cultured for 4 days in a 37 °C humidified chamber at 5% CO_2_ as described [29]. They were cultured in serum-free α-MEM (minimum essential medium eagle) medium (Invitrogen, Burlington, ON, Canada, #12571-063) containing 0.2% bovine serum albumin (Fisher, Ottawa, ON, Canada, #BP1600-100), 0.25% L-glutamine (Invitrogen, Burlington, ON, Canada, #25030-081), 0.216 mg/mL β-glycerophosphate (Sigma, Oakville, ON, Canada, #G9891), 0.05 mg/ml ascorbic acid (Sigma, Oakville, ON, Canada, #A4034), and 0.4% penicillin–streptomycin (Invitrogen, Burlington, ON, Canada, #15140-122) as described [29]. On days 1 and 3, the medium was replaced, and the bones were treated with DMSO (vehicle) control or SR11237 at 3 increasing concentrations (0.1, 1, and 5 μM). Total bone length was determined immediately following isolation and upon experimental completion on day 4, using a Leica S6 D microscope with EC3 camera and Leica Application Suite version 3.8.0 software (Leica Microsystems Inc., Concord, ON, Canada). Following fixation in 4% paraformaldehyde overnight at room temperature, the tibiae were prepared for paraffin embedding and sectioning. The sections were stained with Safranin O/Fast Green as described above.

### 4.4. Micro-Computed Tomography (MicroCT)

Following harvest of the P16 rats, the left limbs were removed, fixed overnight in 4% paraformaldehyde (Sigma, Oakville, ON, Canada, #P6148), and stored in 70% ethanol at 4 °C until scanning. Males were scanned using the GE eXplore speCZT at a resolution of 50 micrometers/voxel and analyzed using the Microview 2.5.0-3943 software (Parallax Innovations, Ilderton, ON, Canada) [30].

### 4.5. Histology and Immunohistochemistry

Humerus, radius, tibia, and femur were harvested and isolated from the right limbs of all animals. Long bones were measured, fixed at room temperature for 24 hours with 4% paraformaldehyde (Sigma, Oakville, ON, Canada, #P6148), and decalcified for 12 days with 5% ethylene diamine tetra-acetic acid (Sigma, Oakville, ON, Canada, #EDT001.5) in phosphate-buffered saline. Bones from rats and mouse ex vivo cultures were processed, embedded in paraffin wax, and sectioned by the Molecular Pathology Laboratory at Robarts Research Institute (London, ON, Canada) at a thickness of 5 μm. Sections were dewaxed using xylenes and then rehydrated by a series of graded ethanol. Following rehydration, the sections were chemically stained, or immunohistochemistry was performed. The sections were stained with 1.5% Safranin O (Sigma, Oakville, ON, Canada, #S8884)/0.01% Fast green (Harleco (Millipore), Etibocoke, ON, Canada, #210-12) as described [31,32,33], picro-sirius red (Picric acid saturated solution Sigma, Oakville, ON, Canada, #P6744-1GA and Sirius Red F3B Sigma, Oakville, ON, Canada, #36-554-8) as described [30], tartrate-resistant acid phosphatase (TRAP) (Sigma, Oakville, ON, Canada, #387-A) according to the manufacturer’s instructions with minor changes (60 minute Triton-X incubation), and terminal deoxynucleotidyl transferase dUTP nick-end labeling (TUNEL) (Calbiochem (Millipore), Etobicoke, ON, Canada, #QIA33) as instructed by the manufacturer’s manual. Immunohistochemistry was performed as described [31] with primary antibodies against PCNA (proliferating cell nuclear antigen) (Cell Signaling, Danvers, MA, USA, #2586) at a concentration of 1:5000, P57 (Santa Cruz, Dallas, Texas, USA, #sc-8298) at a concentration of 1:200, or SOX9 (R&D, Minneapolis, MN, USA, #AF3075) at a concentration of 1:300. Antigen retrieval was completed using 0.1% Triton-X in water, and slides without primary antibody were used as negative controls. All antibody incubations were performed overnight at 4 °C. After washing, secondary antibodies conjugated to horseradish peroxidase (either Santa Cruz, Dallas, TX, USA, #sc-2004 or sc-2020) were used for 1 hour at room temperature, followed by colorimetric detection by diaminobenzidine substrate (Dako, Santa Clara, CA, USA, #K3468). Following this, the sections were counterstained using methyl green (Sigma, Oakville, ON, Canada, #198080). A total of 6 sections per sample were used, with at least 3 independent samples (animals) for each assay. Images were captured directly with a Leica DM1000 microscope and Leica DFC295 camera or stitched together from multiple individual images using Leica Application Suite software.

### 4.6. Statistical Analyses

GraphPad Prism software version 6 (GraphPad Software Inc., San Diego, CA, USA) was used to conduct all statistical analysis. Animals weights and bone lengths were analyzed using a non-parametric Mann–Whitney unpaired two-tailed *t*-test with five independent trials. Organ culture tibial lengths were analyzed using a Kruskal–Wallis one-way ANOVA with five independent trials with Dunn’s post-hoc test. Significance was assumed for p values less than 0.05 for all statistical tests.

## Figures and Tables

**Figure 1 ijms-20-05198-f001:**
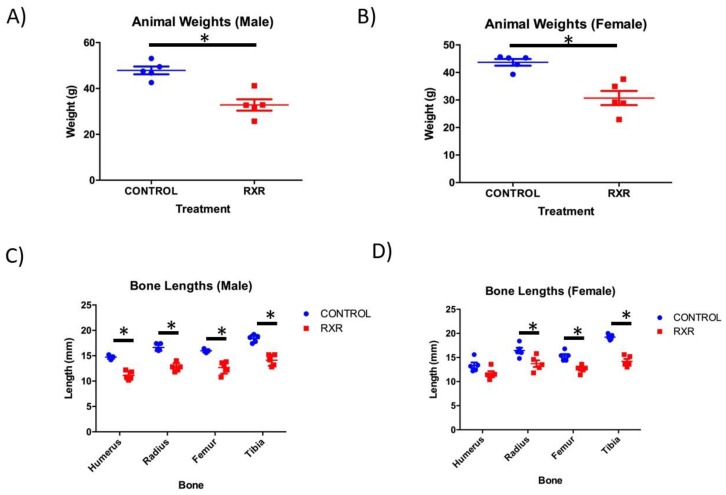
Retinoid X Receptor (RXR) agonist treatment results in reduced weight and bone length in rats. At P16, male (**A**) and female (**B**) animals were weighed before sacrifice (*N* = 5; mean ± SEM; Mann–Whitney unpaired two tailed *t*-test; *p* < 0.005). Bone lengths were measured at P16 in male (**C**) and female (**D**) rats following sacrifice. All RXR agonist-treated bones in the males were found to be significantly shorter than control bones, and in the females, the femur, tibia, and radius treated with the RXR agonist were significantly shorter than the control bones (*N* = 5; mean ± SEM; Mann–Whitney unpaired two-tailed *t*-test; * *p* < 0.005).

**Figure 2 ijms-20-05198-f002:**
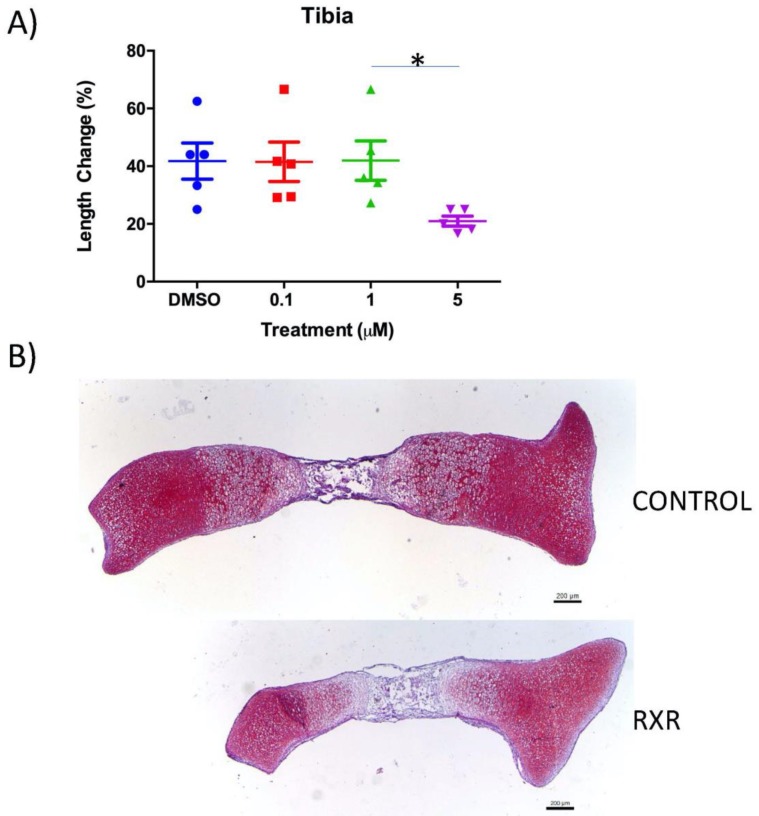
The RXR agonist SR 11237 decreases bone growth in murine tibiae in vitro. P0 tibiae were isolated and cultured for 4 days with DMSO and various concentrations of SR11237. The total length of bones was measured following isolation and upon experimental completion to determine the percentage of longitudinal growth. Treatment of tibia with 5 μM SR11237 caused a decrease in growth compare to all other conditions, but significance was only observed in comparison to the 1 μM treatment (**A**) (*N* = 5; mean ± SEM; Kruskal–Wallis one-way ANOVA with Dunn’s post-hoc test; * *p* < 0.005). The DMSO -treated tibia (top) is trending to be longer than the tibia treated with 5 μM SR11237 (bottom) (**B**).

**Figure 3 ijms-20-05198-f003:**
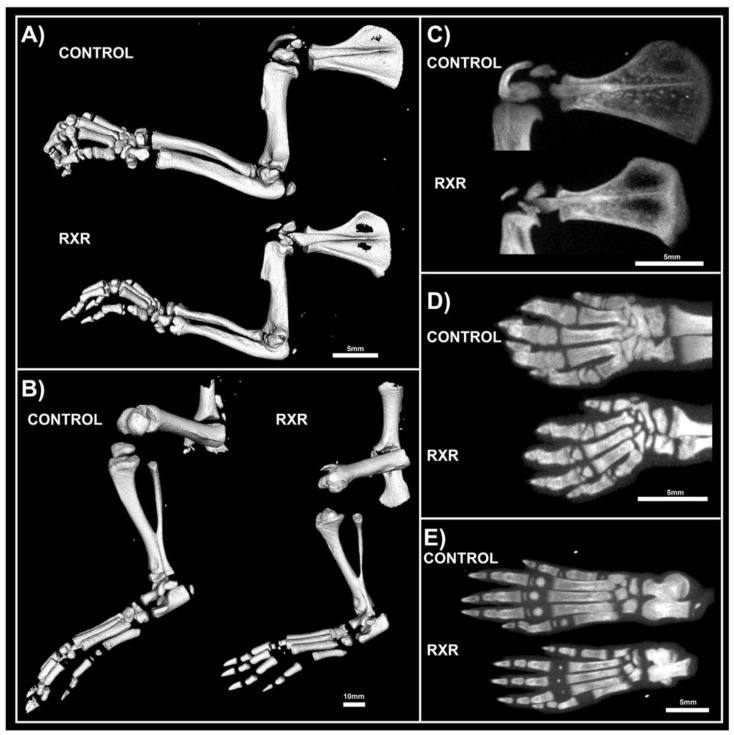
microCT (µCT) Images of P16 RXR and control male rats show abnormal morphology. Fore- and hindlimbs bones of control rats were longer and thicker than those of RXR agonist-treated animals (**A**,**B**). The scapulae of RXR agonist-treated animals had increased radio-opacity in the center of the bone, but more calcification along the outer edges when compared to those of the animals treated with DMSO (**C**). The metacarpals and metatarsals of the hands and feet appeared dysmorphic, under-calcified, and under-developed in the animals treated with the RXR agonist (**D**,**E**) (*N* = 5; 50 micrometers/voxel).

**Figure 4 ijms-20-05198-f004:**
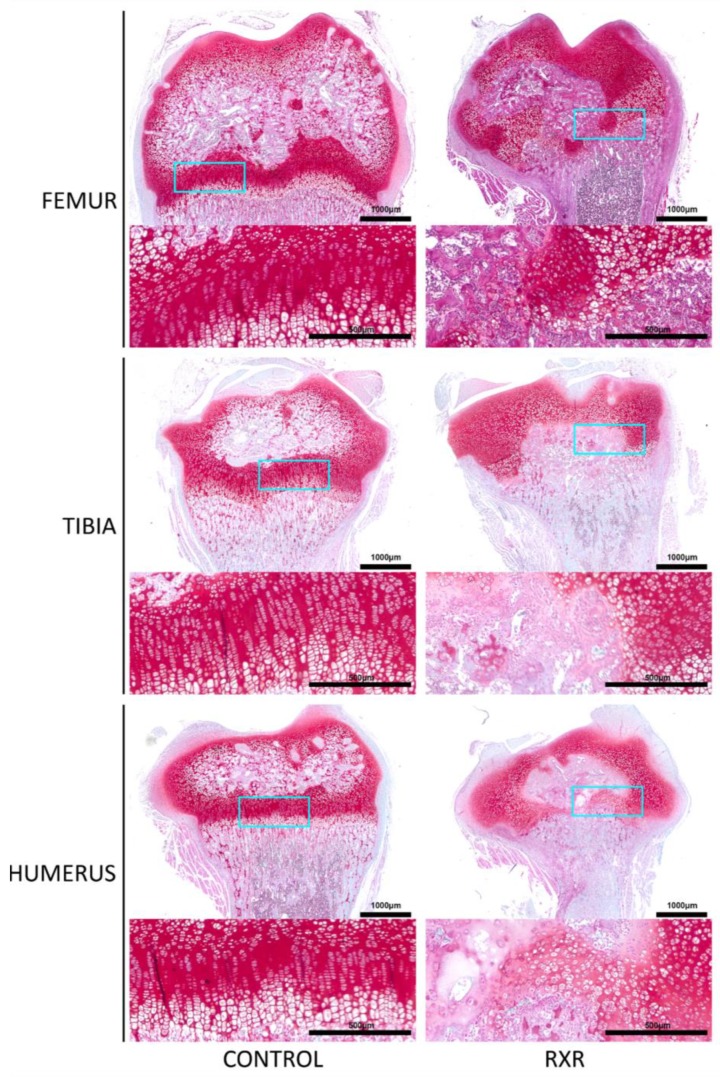
Disrupted growth plate morphology in P16 male rat long bones. Safranin O/fast green staining of bone sections highlights the appearance of disturbed growth plate organization and fusion of primary and secondary ossification centers in RXR agonist-treated males. Higher magnification in inset (scale bar = 1000 μm; inset = 500 μm).

**Figure 5 ijms-20-05198-f005:**
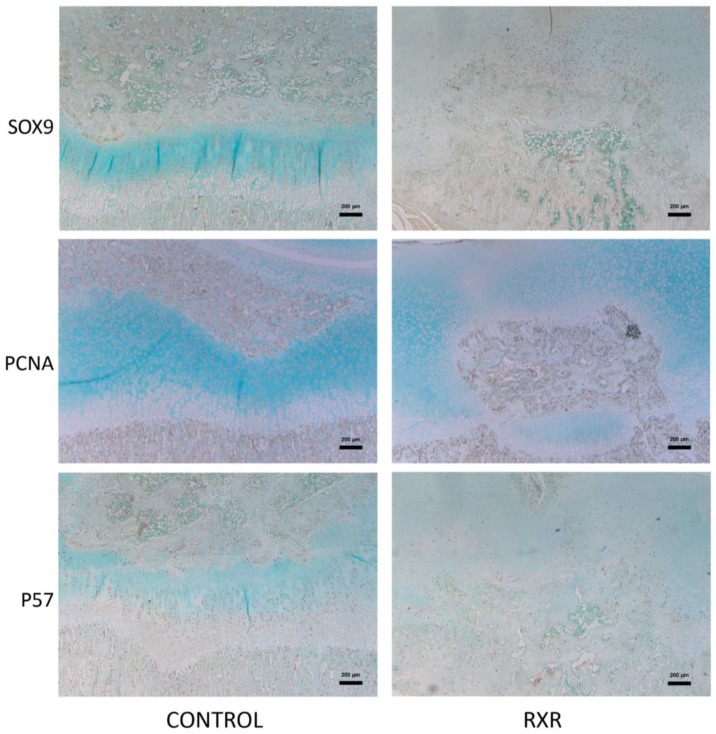
Immunohistochemistry staining of P16 rat tibial sections. Sections of tibiae from rats treated with DMSO or SR11237 were examined by immunohistochemistry for various markers. PCNA is a proliferative marker; P57 demonstrates the arrangement of terminally differentiating chondrocytes in the hypertrophic region; SOX9 shows the organization of proliferating chondrocytes (scale bar = 200 μm).

**Figure 6 ijms-20-05198-f006:**
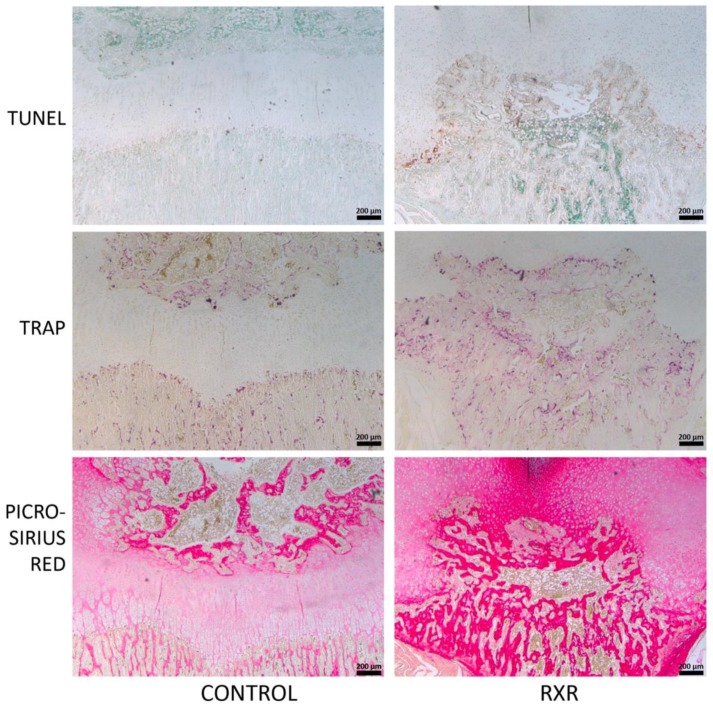
Histological staining of P16 rat tibial sections. Sections of tibiae from rats treated with DMSO or SR11237 were examined by staining for various markers. TUNEL detects cell death; active osteoclasts are stained by TRAP; picro-sirius red stains collagen fibers (scale bar = 200 μm).

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
