# Peer review of "Exposure to the RXR Agonist SR11237 in Early Life Causes Disturbed Skeletal Morphogenesis in a Rat Model"

_ijms, 2019, doi:10.3390/ijms20205198_

Round 1
Reviewer 1 Report
To reveal the function of retinoid x receptor (RXR) in endochondral ossification, the authors investigated the effect of RXR agonist SR11237 on rat endochondral bone development. Although the mechanism of the bone development regulated by RXR is still not clarified, they observed that the activation of RXR causes the disruption of ossification. In this manuscript, the authors well discuss the obtained results and provide useful information for the elucidation of role of RXR in bone development.
Minor comments:
In page 1, line 28, is “RXR receptor” “RXR”?Author Response
Comments: In page 1, line 28, is “RXR receptor” “RXR”?
Response: Thanks for the feedback. Yes, this is what we meant. It has been edited in the paper.
Reviewer 2 Report
In the manuscript entitled “Exposure to the RXR agonist SR11237 in early life causes disturbed skeletal morphogenesis in a rat model” by Dupuis et al. address a relevant point of not questioned so far, that is the effect of the activation of RXR using SR11237 as agonist during bone formation. The manuscript is well written, the data is clear and address original hypothesis. The findings are extremely interesting and will open a new field of the role of RXR.
There are minor observations that need to be address before final acceptance. However, the authors should fully address the points 1 and 2.
Is the SR11237 specific for RXR homodimer or it can activate RXR-RAR or another RXR heterodimer? Did the Authors measure the density of the bones? The SR11237-induced abnormal bone morphology would represent an abnormal bone formation in the adult rat and mouse and thus possible problems, such as walking? On figure 2B, it seems that the Tibiae in vitro treated with SR11237 was not fully extended and somehow folded on the left side. Would the Authors have another picture were the Sr11237 treated Tibiae was more straight to fully show the difference observed in the statistical analysis? The Authors should clarify the thickness of the slice. What is the reason to use only on figure 2 murine data set and the other rat data set? This point should be clarify in the Discussion. Line 175, the Authors state “This study examines the effects of RXR signaling on rodent endochondral ossification and growth plate biology.” However, the data set did not support for a study the signaling itself, rather the effect of administration of SR11237. The Authors should correct this point. Line 257, The Authors should provide the information of how the rats were anesthetized and sacrificed. Line 270, The Authors should provide the amount and concentration of injected DMSO. Line 289, Up to this point there is no description of how the Safranin O- Fast Green staining was performed. The Authors should correct this point. Line 330, The Authors should clarify whether the data pass the homogeneity of variance and provide the F ANOVA with the degrees of freedom. In case that the data did not pass the homogeneity of variance, the Authors should consider to apply a non-parametric test.
Author Response
Question:
Is the SR11237 specific for RXR homodimer or it can activate RXR-RAR or another RXR heterodimer?
Answer:
SR11237 is a specific pan agonist, which means it activates all isoforms of RXR. Thus, it is likely that this compound can activate all permissive heterodimers of RXR (as discussed in the introduction), although this has not been formally shown to our knowledge. The literature states that SR11237 is completely devoid of RAR activity; in agreement with that, it should not activate any non-permissive heterodimers.
Question:
Did the Authors measure the density of the bones?
Answer:
Although quantitative measurements of the density of the bones would have been ideal, the resolution of the uCT did not allow for this assessment. Please also note that our focus was on the growth plate, which appeared to be the primary skeletal tissue affected, rather than bone itself.
Question:
The SR11237-induced abnormal bone morphology would represent an abnormal bone formation in the adult rat and mouse and thus possible problems, such as walking?
Answer:
The animals were harvested immediately upon completing the injections with SR11237. Although a behavioural assessment (such as Catwalk gait analyses) would have been interesting to assess the animal’s ability to move, such assays have not been established for this age, to the best of our knowledge.
Question:
On figure 2B, it seems that the Tibiae in vitro treated with SR11237 was not fully extended and somehow folded on the left side. Would the Authors have another picture were the Sr11237 treated Tibiae was more straight to fully show the difference observed in the statistical analysis?
Answer:
Measurements were done on intact ex-vivo bones before histological processing. Great lengths were taken to ensure the bones were as straight as possible during measurement. Tissues were processed for sectioning, and bends in the tissue may have been exacerbated during this process. Unfortunately, we do not have any images that are straighter for publication.
Question:
The Authors should clarify the thickness of the slice.
Answer:
All sections were cut at 5um. This value is communicated in the methods section (line 344).
Question:
What is the reason to use only on figure 2 murine data set and the other rat data set? This point should be clarify in the Discussion.
Answer:
The organ culture system is only established in mouse in our laboratory. We also believe that the conclusions of our study are strengthened by finding similar effects in two species. A section has been added in the discussion to address this (lines 218-226).
Question:
Line 175, the Authors state “This study examines the effects of RXR signaling on rodent endochondral ossification and growth plate biology.” However, the data set did not support for a study the signaling itself, rather the effect of administration of SR11237. The Authors should correct this point.
Answer:
Thank you for the suggestion. The changes have been reflected throughout the manuscript, e.g. in the abstract (line 27), discussion (line 201), and conclusions (line 288).
Question:
Line 257, The Authors should provide the information of how the rats were anesthetized and sacrificed.
Answer:
The animals were asphyxiated by CO2. This clarification has been added to the materials and methods (line 299, 301).
Question:
Line 270, The Authors should provide the amount and concentration of injected DMSO.
Answer:
The animals were injected with 25mg/kg of SR11237 diluted in DMSO. The control animals were injected with the same volume of DMSO. This information was changed in the materials and methods section (line 309).
Question:
Line 289, Up to this point there is no description of how the Safranin O- Fast Green staining was performed. The Authors should correct this point.
Answer:
The protocol for the Safranin O / Fast green staining was explained in the material and methods section Line 346), with references to our published work.
Question:
Line 330, The Authors should clarify whether the data pass the homogeneity of variance and provide the F ANOVA with the degrees of freedom. In case that the data did not pass the homogeneity of variance, the Authors should consider to apply a non-parametric test.
Answer:
Thank you for pointing this out. Upon further review with the comments in mind, statistical analyses were re-done using non-parametric methods. The results of this change are reflected in the manuscript (e.g. Figures 1 and 2). The main conclusions that SR11237 affects bone growth was not altered, although the effects in the mouse ex vivo culture were less pronounced. This is discussed (lines 218-226).